# Comparisons of Anatomical Characteristics and Transcriptomic Differences between Heterografts and Homografts in *Pyrus* L.

**DOI:** 10.3390/plants11050580

**Published:** 2022-02-22

**Authors:** Piyu Ji, Chenglin Liang, Yingjie Yang, Ran Wang, Yue Wang, Meitong Yuan, Zhiyun Qiu, Yuanyuan Cheng, Jianlong Liu, Dingli Li

**Affiliations:** 1Qingdao Key Laboratory of Genetic Improvement and Breeding in Horticultural Plants, Engineering Laboratory of Genetic Improvement of Horticultural Crops of Shandong Province, College of Horticulture, Qingdao Agricultural University, Qingdao 266109, China; jipiyu123@163.com (P.J.); wuhuaguoyyj@163.com (Y.Y.); qauwr@126.com (R.W.); lhqxxwy@163.com (Y.W.); yuanmt123@126.com (M.Y.); qauqzy2020@163.com (Z.Q.); chengyy0923@163.com (Y.C.); 201901068@qau.edu.cn (J.L.); 2Haidu College, Qingdao Agricultural University, Laiyang 265200, China; lcliangchenglin@163.com

**Keywords:** pear, grafting, graft union, RNA sequencing, β-glucosidase

## Abstract

Pear (*Pyrus* L.) is an important temperate fruit worldwide, and grafting is widely used in pear vegetative propagation. However, the mechanisms of graft healing or incompatibility remain poorly understood in *Pyrus*. To study the differences in graft healing in *Pyrus*, the homograft “Qingzhen D1/Qingzhen D1” and the heterograft “QAUP-1/Qingzhen D1” as compatibility and incompatibility combinations were compared. Anatomical differences indicated the healing process was faster in homografts than in heterografts. During the healing process, four critical stages in graft union formation were identified in the two types of grafts. The expression of the genes associated with hormone signaling (auxin and cytokinins), and lignin biosynthesis was delayed in the healing process of heterografts. In addition, the *PbBglu13* gene, encoded β-glucosidase, was more highly up-regulated in heterografts than in homografts to promote healing. Meanwhile, the most of DEGs related starch and sucrose metabolism were found to be up-regulated in heterografts; those results indicated that cellulose and sugar signals were also involved in graft healing. The results of this study improved the understanding of the differences in the mechanisms of graft healing between homografts and heterografts.

## 1. Introduction

Pear is one of the most important deciduous fruit crops worldwide, and it has been cultivated for 2000 years in the world [1]. Grafting is the main methods of pear vegetative propagation, by mixing the characteristics of rootstock and scion, and then the graft plants can achieve a stronger tolerance or resistance to biotic or abiotic stresses, dwarfing traits, and a higher yield [2,3]. However, the process and molecular mechanism of graft healing remain poorly understood.

The process of graft healing mainly includes callus formation, cambium connection and differentiation, and establishing a connection between the xylem and vascular bundle [4]. After grafting, ruptured cells collapse and adhere to opposing tissue at the graft junction, where a mass of pluripotent cells called the callus are formed [5]. The callus is formed on both sides of the junction until the grafted tissues join, and then the plasmodesmata bridges the junction and ultimately reconstruct vascular bundles [6]. Although there might be great differences in grafting between different species, the connective tissue between rootstock and scion is the basis of successful grafting [7]. However, grafting incompatibility often occurs in heterografts, leading to low rates of survival, abnormal growth, and physiological disorders [8,9,10,11,12,13,14,15]. Graft incompatibility occurs in a variety of horticultural species. When cucumber is grafted onto incompatible pumpkin rootstocks, its necrotic layer disappears later than when grafted onto compatible rootstocks [15,16,17]. When *Arabidopsis* is grafted onto cabbage, radish, and tobacco, although callus forms at the graft junction, there is a few complete vascular bundles [14,18]. Meanwhile, heterografts and homografts are usually used as a model system to determine incompatible and compatible combinations of graft union [19,20]. In litchi, vascular bundles of heterografts reconnect normally in the early stage, but in the later stage, gaps formation can be observed [19,21]. When pepper and tomato are grafted, parenchymatous callus forms at the junctions of compatible and incompatible combinations, but in the incompatible combinations, abnormal vascular tissue and xylem discontinuity are observed [20]. 

The process of graft healing involves complex signal transduction and material accumulation. Plant hormones, including auxins, cytokinins (CKs), ethylene, gibberellins, and jasmonic acid, are important in growth and development and also in the graft healing process, in which they affect cell division and differentiation, wound healing, and vascular formation [22,23]. Auxin activates *ALF4* and *AXR1* in order to promote graft formation, and auxin appears to be the primary regulator of vascular cell differentiation [24]. Other plant hormones such as cytokinins and gibberellins interact with auxin biosynthesis via transport and signaling pathways, and thereby also function in the graft healing process [25]. In previous studies, some genes related hormone synthesis and signal transduction were identified, and the expression patterns of those genes were different in the healing process of different grafting combinations. The genes related plant hormone synthesis and signal transduction respond faster in homografts or compatible combinations [15,20,21]. During plant growth and development, lignin, cellulose, and hemicellulose are important substances in cell wall formation [26]. Following the completion of cell elongation, hemicellulose and lignin are deposited in secondary cell walls to increase cell wall strength and thickness [26,27]. Genes associated with lignin and cellulose biosynthesis are up-regulated in graft healing in litchi and pecan [19,28]. In different grafting combinations, lignin content and synthesis related genes show great differences at different stages in the grafting healing process [7,8,19]. Furthermore, sugar also has an important role in graft junction formation, and sugar response, ranging from asymmetric to symmetric, is the main event in graft development [29]. In nonfunctional grafts of tomato, accumulations of starch and soluble sugar also increase in the scion [30]. Several cellular sugar transporters have been identified that affect the graft healing process by participating in sucrose efflux through cell membranes [31,32]. In *Arabidopsis thaliana*, compared with no addition, the addition of sugar to the grafting medium increases the graft success rate and speeds up recovery after grafting [33]. Sugars also promote graft union development in heterografts of cucumber onto pumpkin [16]. Asymmetric accumulation of sugar in rootstock and scion is a more serious problem in incompatible combinations. In the scion of melon grafted onto incompatible rootstocks, starch accumulation increases [11]. In the period of graft union formation, substantial changes occur in cell differentiation and organ regeneration, and there is an even exchange of genetic materials [34,35]. With the development and application of gene sequencing technology in plants, transcriptomic analysis had provided key information about genes associated with vascular connections between rootstock and scion by identifying differentially expressed genes (DEGs) between compatible and incompatible graft combinations [11,36,37].

Grafting has been widely used to enhance the productivity and resilience of fruit corps, but graft healing and incompatibility confine the technique to further utilization. There are few reports in *Pyrus*, so homografts and heterografts as the compatibility and incompatibility combinations were selected here for exploring the mechanisms of graft in *Pyrus*.

## 2. Results

### 2.1. Effects of Homografts and Heterografts on Grafting Junction

Stem diameters of homografted and heterografted one-year-old clonal plants were measured one year after grafting. When other parts of the stem were compared with the stem 4 cm above the grafting junction, the stem diameter ratio of the grafting union in homografts was significantly greater than that in the heterografts (Figure 1a). Analysis of the mineral element contents in different parts of stems after grafting showed that the calcium (Ca) content was higher in the stems at the grafting union in the homografts than in the heterografts, whereas the phosphorus (P), potassium (K), and magnesium (Mg) contents were higher in the stems above the grafting union in the heterografts than in the homografts (Table 1). To measure the binding force between the rootstock and scion in the graft healing process of homografts and heterografts, plantlets of “Qingzhen D1” and “QAUP-1*”* were used as rootstocks and scions, and were micrografted with silicon tubings. The binding force gradually increased and was significantly different between homografts and heterografts at 15 and 30 DAG (Figure 1b), and had a greater binding force; this result suggests that the union in homografts perhaps formed faster than that in heterografts.

### 2.2. Anatomical Observations in the Healing Process of Homografts and Heterografts

To determine differences in the formation of grafting unions, the anatomical structure of homografts and heterografts was examined at 1, 5, 9, 25, and 30 DAG (Figure 2). In the homografts, callus tissue filled the interface spaces between the scion and rootstock at 5 DAG, but in the heterografts, a necrotic layer remained at the graft interface. At 9 DAG, parenchymatous cells close to the graft interface proliferated and began to differentiate. However, at the same stage, callus tissue appeared on the rootstock and scion. At 25 and 30 DAG, new vascular tissues formed to connect the xylem and phloem between the scion and rootstock in the homografts (Figure 2). In heterografts, vascular connection and development were abnormal, and gaps were still observed at 25 and 30 DAG (Figure 2). On the basis of anatomical observations, the homografts had a higher survival rate and a faster healing process than those of the heterografts. The observations indicated that 1, 5, 9, and 30 DAG were the critical stages, which were named the SS (separation of rootstock and scion), CF (callus formation), CD (cambium differentiation), and VC (vascular connection) stages, respectively. 

### 2.3. Comparison of Transcriptional Profiles between Homografts and Heterografts in the Healing Process of Graft Unions

To explore the mechanisms underlying the differences in the healing process between homografts and heterografts, global transcriptional profiles of different stages in the healing process between homografts and heterografts sampled from plantlets of “Qingzhen D1” and “QAUP1” were analyzed. In homografts, there were four critical stages in the healing process, namely SSS, SCF, SCD, and SVC, whereas in heterografts, they were DSS, DCF, DCD, and DVC. Overall, all libraries had good sequencing quality with Q30 exceeding 94.22% (Appendix A). As shown in Appendix A, heterograft and homograft samples clustered separately in the first clade, and the three replicates for each treatment clustered together in the final clade, indicating that the gene expression profiles for the samples and replicates were highly consistent. We randomly selected 15 genes to analyse in qPCR in order to provide an independent assessment of the reliability of RNA-seq data. The high correlation coefficient (R2 = 0.7088) between qPCR and RNA-seq results indicates that the RNA-seq data were reliable (Appendix A). A total of 1874 DEGs were identified in the DSS vs. SSS comparison, with 1114 up-regulated and 760 down-regulated; 796 DEGs were identified in the DCF vs. SCF comparison, with 475 up-regulated and 321 down-regulated; 276 DEGs were identified in the DCD vs. SCD comparison, with 190 up-regulated and 86 down-regulated; and 370 DEGs were identified in the DVC vs. SVC comparison, with 309 up-regulated and 61 down-regulated (Appendix A).

### 2.4. Pathway Enrichment Analysis at Different Stages of Junction Formation

A Kyoto Encyclopedia of Genes and Genomes (KEGG) enrichment analysis was performed to identify the relevant metabolic pathways in which DEGs participated. To study the graft healing process, KEGG pathways enriched with DEGs were examined in the homografts. Among the enriched pathways, the “phenylpropanoid biosynthesis” pathway was identified in three comparisons, which was consistent with the significant role of this metabolic pathway during the grafting process (Figure 3). The “phenylalanine metabolism” pathway was also an important secondary metabolic pathway and was enriched with DEGs in three stages of graft formation. Downstream of this pathway was primarily divided into flavonoid and lignin biosynthesis pathways, which produce phenylpropane metabolites, including coumarins, flavonols, lignin, cork esters, and other benzene compounds (Figure 3a). The “phenylpropanoid biosynthesis” and “plant hormone signal transduction” pathways were particularly enriched in the DCF vs. SCF, DCD vs. SCD, and DVC vs. SVC comparisons (Figure 3b*,*c).

### 2.5. Hormonal Signaling Was Altered in the Healing Process of Graft Unions of Heterografts

Plant hormones are necessary for growth and development. To explore roles of hormone signal transduction in the graft healing process in *Pyrus*, further KEGG pathway enrichment analysis was conducted. Differentially expressed genes were primarily enriched in the auxin and cytokinin signal transduction pathways. Differentially expressed genes involved in auxin signaling included AUX1 and AUX/IAA. The genes were highly expressed in the early stages of homografts, but the expression was low in the later stages of heterografts. The expression of auxin-responsive GH3 and SAUR was delayed in homografts compared with that in homografts (Figure 4a). In cytokinin signal transduction, the expression patterns of cytokinin-related factors type-B ARR were similar to those genes of the auxin signal transduction pathways, and had a high expression at the early stages in the homografts. Type-A ARR transcription factors, which negatively regulate the cytokinin signaling pathway, showed the opposite trend in the grafting healing process. (Figure 4b).

### 2.6. Lignin Biosynthesis in Graft Healing Process

Lignin biosynthesis is important in forming the vascular bundle, and successful grafting in plants requires a functional vascular system between the scion and rootstock. Phenylalanine was successively catalyzed by phenylalanine ammonia-lyase (PAL), 4-coumarate−CoA ligase (4CL), cinnamoyl-CoA reductase (CCR), cinnamyl-alcohol dehydrogenase (CAD), and lignin peroxidase (POD) to form lignin. In this study, DEGs encoding enzymes in the general phenylpropanoid pathway were identified. As shown in Figure 5, the DEGs in the lignin biosynthesis pathway included one gene encoding Pb4CL, four encoding *PbCCR*, and five encoding *PbCAD*. The expression of DEGs tended to increase during the graft healing process and indicated that more genes might have roles in the early stages of healing in the homografts than in the heterografts.

### 2.7. Activity of β−Glucosidase Was Higher in Heterografts than in Homografts

On the basis of shared DEGs in roots in the comparisons of SSS vs. DSS, SCF vs. DCF, SCD vs. DCD, and SVC vs. DVC, 40 genes were influenced by heterografts in the graft healing process, compared with homografts (Figure 6a). The KEGG enrichment analysis indicated that 10 pathways were the most significantly enriched in 40 DEGs, including the “starch and sucrose metabolism” pathway (Figure 6a), which might help explain differences in the healing process between homografts and heterografts. A gene encoding β−glucosidase, PbBglu13, was more highly up-regulated in heterografts than in homografts in the healing process (Figure 6b). The highest activity of β−glucosidase was in the stage CF in heterografts, and activity was significantly higher in stage CF, CD, and VC in heterografts than in homografts (Figure 6c). After β-glucosidase treatment, vascular bundle connection was completed at the 10 DAG, which was significantly earlier than that in the untreated group (Figure 6d). To study the possible role of sugar metabolism in graft healing of two combinations, we analyzed DEGs enriched in the “starch and sucrose metabolism” pathway. Those DEGs were encoding trehalose 6−phosphate phosphatase (ostB), β−glucosidase (Bglu), invertase (Inv), β−amylase (Bmy), endoglucanase (Eglu), glycogen phosphoylase (Pyg), and sucrose synthase (Sus). Most of the DEGs encoding ostB, Bmy, Sus, and Pyg were highly expressed in the healing process of heterografts compared with homografts (Figure 7). There were a few genes that were up-regulated in the early stages in homgrafts compared with in heterografts (Figure 7). 

## 3. Discussion

Grafting technology has a long history in pear production. Graft healing is a complex development process that is affected by many factors, including the temperature, humidity, grafting method, and genotype. Suitable environment and grafting methods could improve the grafting survival rate to a certain extent [38,39], but the genotypes of the scion and rootstock were the key factor affecting the graft healing and incompatibility. In this study, the heterografts and the homografts were used as a model system to study the difference between compatible and incompatible combinations. Graft junctions showed greater swelling after one year in heterografts than in homografts (Figure 1a). Calcium is an important element in plant growth and development, and it affects cell wall formation by regulating lignin and other secondary metabolites [40]. The calcium content in the stems at the grafting junction was higher in the homografts than in heterografts, suggesting grafting junctions were abnormal in the heterografts. Micrografting protocols have been used widely in plants to study signal transduction and material communication in grafted plants [31,41]. In this study, a micrografting technique was used to investigate the healing process in homogeneous and heterogeneous grafting combinations. Anatomical observations and the binding force of scions and rootstocks indicated the process of the grafting union was different between the two combinations (Figure 1a). To identify differences at the molecular level, transcriptome analysis was conducted to compare the DEGs in four critical stages of graft union development in homografts and heterografts.

Plant hormones are likely required for graft union formation. In this study, hormone-related genes were primarily enriched in auxin and cytokinin signal pathways. Auxin is involved in vascular formation [24], and the expression of DEGs associated with the auxin pathway is high 2 h after grafting in litchi [20], which indicates that IAA has a role in the early stage of graft healing. In this study, a gene encoding an auxin influx carrier had a high level of expression beginning in stage SS in the homografts, whereas in heterografts, the expression level of the gene was high beginning in stage CF. In addition, the expression of auxin-responsive genes (*AUX/IAA*, *GH3*, and *SUAR*) was up-regulated in later stages of graft healing in heterografts compared with that in homografts, with only a few of the genes highly expressed at stages SS and CF in the homografts. This result demonstrated that activation of the IAA hormone signal might be faster in homografts than in heterografts. Cytokinins are associated with cell division, including cambium cell division [42,43,44,45]. Cytokinin signals are mediated via a two-component regulatory pathway that activates type-B ARR and type-A ARR transcription factors, which negatively regulate the cytokinin signaling pathway [46]. In this study, one gene encoding type-B ARR was more highly expressed after grafting, beginning in stage SS in homografts; however, the same was not observed in heterografts. In addition, the expression of type-A ARRs increased significantly during late graft stages in homografts; however, the expression level in the incompatible combination showed high trends. These results indicated that the expression of auxin and cytokinin signals was faster in the homografts than in the heterografts.

Lignin is a primary component of secondary cell walls and is important in the formation of vascular bundles [26,27]. The genes *4CL*, *C**AD*, and *POD* encode key enzymes in lignin biosynthesis [19,47], and in previous studies, the up-regulated expression of those genes promoted lignin synthesis. In the graft healing process of *Litchi chinensis*, *Carya cathayensis*, and *Prunus*, the gene expression patterns are similar [19,26,47]. In the homografts in this study, the expression of *Pb4CL*, *PbC**AD*s, and *PbPOD*s was up-regulated after grafting. However, in heterografts, the expression of the genes was delayed compared with that in the homografts, which might lead to abnormal formation of vascular tissue. Cellulose is also a major component of plant cell walls and is responsible for oriented cell elongation during growth and development [48]. The gene *PbBglu13* is involved in β-glucosidase synthesis, which is a cellulase that produces glucose by decomposing cellobiose, a hydrolysate of cellulose glucanase [9,49,50]. The expression of *PbBglu13* was higher in heterografts than in homografts in the process of the grafting junction (Figure 6b). The activity of β-glucosidase was higher in homografts than heterografts (Figure 6c). Interestingly, β-glucosidase promotes the connection of vascular bundles, which may be because the heterojunction promotes the transformation process of sugar and cellulose, and finally leads to healing. This shows that the conversion of sugar and cellulose plays an important role in the process of heterografts. Meanwhile, we analyzed the expression pattern of DEGs enriched in the “starch and sucrose metabolism” pathway, and more DEGs were highly expressed in the healing process in heterografts compared with homografts, which indicates that there was a greater accumulation of sugar in heterografts (Figure 7). There has been some research that reported that soluble sugars were accumulated in incompatible graft combinations [51]. In a recently study of pear cultivars grafted on different rootstocks, compared with heterografts, the sugar content was the lowest in homografts [52]. So, the related pathways of sugar metabolism will be the focus of future research.

In conclusion, anatomical observations were used to determine critical stages in the graft healing process in homografts and heterografts. Based on four critical stages, a hypothetical model of graft union development in compatible and incompatible grafts of *Pyrus* was developed (Figure 8). Hormone signals (IAA and cytokinins) respond to the graft healing process and promote cell regeneration and callus induction. In addition, genes associated with lignin biosynthesis promote vascular reconnection between the scion and rootstock. However, in heterografts, the expression of those genes is delayed significantly, which retards the process of graft healing. The gene *PbBglu13*, which regulates cellulose and sugar signals, also affected the process of graft healing. Collectively, the results of this study revealed the physiological and molecular mechanisms underlying the differences in the graft healing process between homografts and heterografts, which will also help to elucidate the mechanisms involved in the graft healing processes of other species.

## 4. Materials and Methods

### 4.1. Plant Materials and Grafting

The pear cultivars used in the grafts were “Qingzhen D1” (*Pyrus communis* L. × *Pyrus bretschneideri* R.) and “QAUP-1” (*Pyrus ussuriensis* Maxim.). In experiment I, one-year-old clonal plants of “Qingzhen D1” and “QAUP-1” planted in the Weifang Regional Experimental Park were used as the scion and rootstock to be grafted by cleft grafting. “Qingzhen D1” and “QAUP-1” were the scions and “Qingzhen D1” was the rootstock. The homograft was “Qingzhen D1” grafted onto “Qingzhen D1”, and the heterograft was “QAUP-1” grafted onto “Qingzhen D1”. In experiment II, the uniform plantlets of “Qingzhen D1” and “QAUP-1” with about 3 cm in length and 2–4 mm in stem diameter were selected and used as the rootstocks and scions, and the grafting combinations and methods were the same as experiment I. A micro-grafting technique was applied in this experiment, and silicon tubing was used to hold the graft partners in position as the graft union was formed, and the grafted plants were grown on 1/2 MS medium with 1.0 mg·L^−1^ 6-BA, 0.3 mg·L^−1^ IBA, 30 g·L^−1^ sucrose, and 7 g·L^−1^ agar in a tissue culture room. The graft unions of plants in experiment II were harvested at 1, 5, 9, and 30 DAG and were immediately transferred to liquid nitrogen before storage at −80 °C until further use. Three biological replicates were performed for each time point. 

### 4.2. Determination of Scion and Rootstock Stems Diameters in Homografts and Heterografts Plants

The scion and rootstock diameters were measured one year after grafting and the plant materials was the same as in experiment I. The stem diameters of the rootstock and scion were measured 4 cm above and below the grafting junction, per 1 cm. The stem diameter ratio was calculated as the stem diameter of different parts of the stem/stem diameter of scion 4 cm above the grafting junction.

### 4.3. Determination of Mineral Elements in Different Parts of Stem in Homografts and Heterografts

The stem samples of experiment I were collected one year after grafting at different locations, which included the grafting junction and 10 cm above and below the grafting junction. Stem samples were dried in a forced-air oven at 85 °C for two weeks and then ground into powder. Samples of 1.0 g were put in digestion tubes, and 10 mL of concentrated nitric acid and 2 mL of concentrated hypochlorous acid were added for digestion. After digestion, the volume was set to 50.00 mL. The contents of the mineral elements were determined with an inductively coupled plasma emission spectrometer (ICP-OES).

### 4.4. Determinations of β-Glucosidase Activity and Binding Force during Graft Healing, and Exogenous β-Glucosidase Treatments

The β-glucosidase content was measured using a β-glucosidase Activity Assay Kit (Cat# BC2560, Solarbio, China, Beijing) following the manufacturer’s protocols. To test the effect of β-glucosidase on grafting, the grafted “QAUP-1”/ “Qingzhen D1” was applied with 5% (*w*/*v*) β-glucosidase at the graft junction. The plants and culture environment was same as experiment II.

### 4.5. Determinations Binding Force during Graft Healing

The binding force between the scion and rootstock at 1, 7, 15, and 30 days after grafting (DAG) was measured using a digital push–pull meter. Binding force was the peak value at the separation of the rootstock and scion. All of the samples were from experiment II.

### 4.6. Paraffin Section Microscopy

All of the samples of experiment II at 1, 5, 9, 25, and 30 DAG were placed in FAA for 1 d and then were dehydrated in an ethanol series (50%, 60%, 70%, 80%, and 95%), with 60 min at each dehydration step. In the last step, samples were placed in 100% ethanol overnight. After decoloration with dimethylbenzene, the samples were embedded in paraffin. The samples were sectioned vertically to 10 μm using a rotary microtome (Leica RM2235, Germany, Wetzlar), and were dewaxed, rehydrated, cleaned, stained with toluidine blue, counterstained with safranin, and then fixed with neutral balata. The sections were examined and photographed using an optical microscope (Leica RM2235).

### 4.7. Library Construction and Transcriptome Sequencing

A total of 1 µg of RNA per sample was prepared for analysis. To generate sequencing libraries, a NEB Next UltraTM RNA Library Prep Kit for Illumina (NEB, Ipswich, MA, USA) was used following the manufacturer’s instructions. Fragments of cDNA 400–500 bp in length were selected, and an AMPure XP system (Beckman Coulter, Beverly, CA, USA) was used to purify the library fragments. cDNA fragments with ligated adaptor molecules on both ends were selectively enriched using an Illumina PCR Primer Cocktail in a 15-cycle PCR reaction. Products were purified (AMPure XP system) and quantified using a Bioanalyzer 2100 system (Agilent, U.S.A, Santa Clara, CA, USA). The sequencing library was sequenced on a NovaSeq 6000 platform (Illumina). Image files of the samples sequenced on the platform were transformed by sequencing platform software into FASTQ format (Raw Data). Cutadapt v1.15 software was used to obtain high-quality sequences (Clean Data) for further analysis. The reference genome was 4. *Pyrus_bretschneideri_*scaffold., which was built by Pear Genome Project and the database version was v1.0. Gene annotation files were downloaded from the genome website. Filtered reads were mapped to the reference genome using HISAT2 v2.0.5. ClusterProfiler 3.4.4 software was used to conduct the KEGG pathway enrichment analysis of DEGs. RNA-seq samples with three biological replicates were collected from graft unions of plantlets harvested at 1, 5, 9, and 30 DAG (Figure 1c). The sequenced raw data in this study were deposited at NCBI (https://www.ncbi.nlm.nih.gov/bioproject/, accessed on 11 December 2021) under accession number GSE190654.

### 4.8. Analysis of Differentially Expressed Genes

HTSeq 0.9.1 statistics were used to compare the read count values of each gene as the original expression of the gene, and then FPKM was used to standardize the expression. Differentially expressed genes were identified in DESeq (1.30.0) according to the following criteria: expression difference multiple |log2 fold change| > 1; significant at *p* < 0.05.

### 4.9. RNA Extraction and Reverse-Transcription Quantitative PCR

The total RNA was extracted and purified using an RNAprep Pure Plant Kit (Tiangen, Beijing, China) according to the manufacturer’s instructions. RNA quality was verified by spectrophotometric measurement on a NanoDrop 2000C instrument (Thermo Fisher Scientific, Waltham, MA, USA). First-strand cDNA was synthesized using a HiScript II first Strand cDNA Synthesis Kit (Vazyme, Beijing, China) according to the manufacturer’s instructions. Reverse-transcription quantitative PCR was conducted using a Roche 480 real-time PCR system (Basil, Switzerland) in standard mode with a FastStart Essential DNA Green Master kit with the following cycling parameters: 95 °C for 5 min and 45 cycles at 95 °C for 15 s, 60 °C for 30 s, and 72 °C for 30 s. All of the reactions were performed in triplicate in a 20-µL volume containing 2 µL of 10-fold diluted cDNA, with the pear Actin gene used as the internal control. All primers are listed in Appendix A.

### 4.10. Statistical Analysis

All the data were the average values of at least three replicates and their standard deviations. The statistic significant differences were determined using Student’s *t*-tests. Data were expressed as the mean ± standard deviation (SD) of three biological replicates. TBtools was used to make heat maps [53].

## Figures and Tables

**Figure 1 plants-11-00580-f001:**
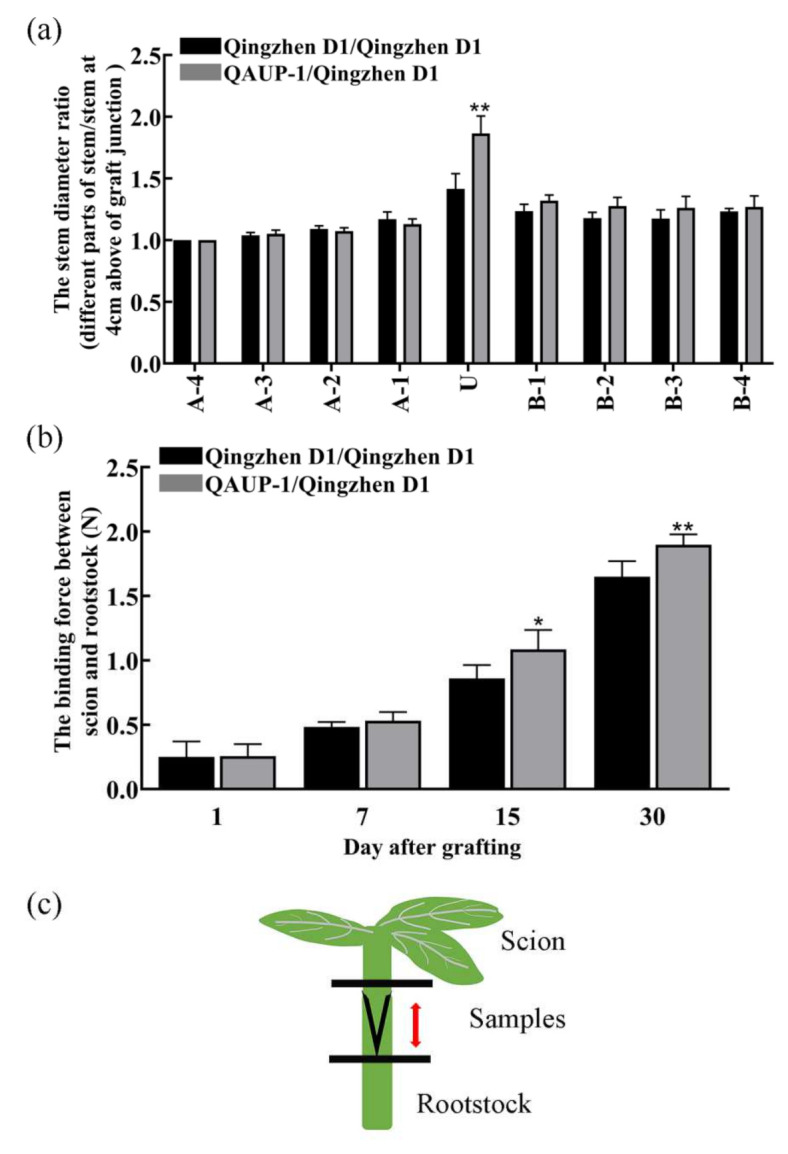
Stem diameter ratio one year after grafting, binding force in the healing process, and sample location in homografts (“Qingzhen D1”/ “Qingzhen D1”) and heterografts (“Qingzhen D1”/”QAUP−1”) of pear. (**a**) Stem diameter ratio (stem diameter of stem part/stem diameter 4 cm above the graft junction) one year after grafting. U: graft junction; A: above graft junction; B: below graft junction. Numbers indicate distance from graft junction in units of 1 cm. (**b**) Binding force in homografts and heterografts at 1, 7, 15, and 30 DAG. Values are the mean ± SE, *n* = 5. Significant differences within a group are expressed as * *p* < 0.05 and ** *p* < 0.01 (Student’s *t*−test). (**c**) Location of samples for RNA sequencing.

**Figure 2 plants-11-00580-f002:**
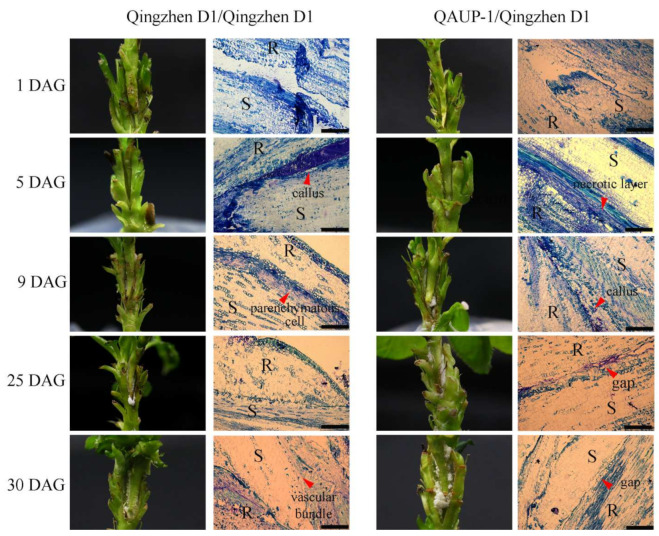
External anatomy (**left**) and histology (**right**) of graft union formation in homografts (“Qingzhen D1”/”Qingzhen D1”) and heterografts (“Qingzhen D1”/”QAUP-1”) of pear. Longitudinal sections of the middle part of graft unions were observed at 1, 5, 9, 25, and 30 days after grafting (DAG) in paraffin sections. S, scion; R, rootstock. Scale bar = 250 μm.

**Figure 3 plants-11-00580-f003:**
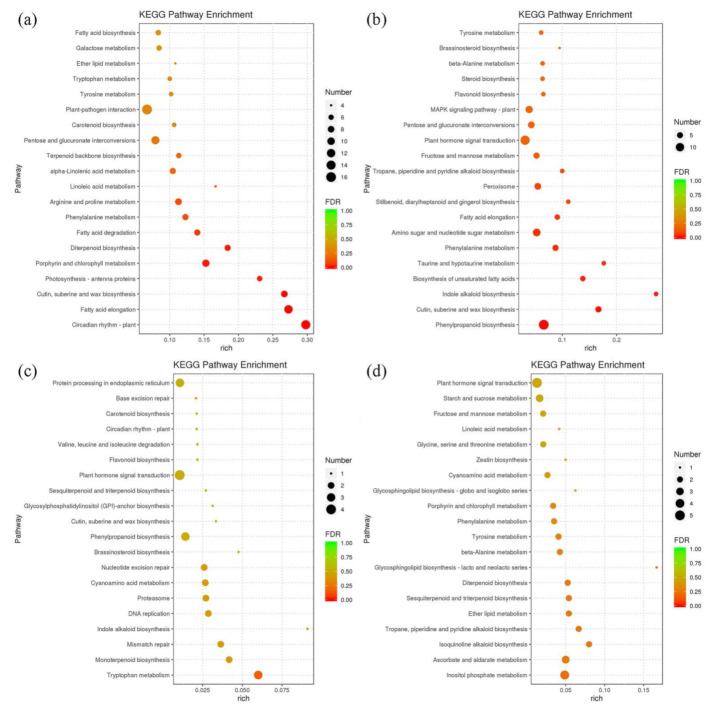
KEGG pathway analysis of differentially expressed genes (DEGs). KEGG terms of DEGs were determined in comparisons of heterografts and homografts at different stages in the graft healing process: (**a**) SSS vs. DSS, (**b**) SCF vs. DCF, (**c**) SCD vs. DCD, and (**d**) SVC vs. DVC.

**Figure 4 plants-11-00580-f004:**
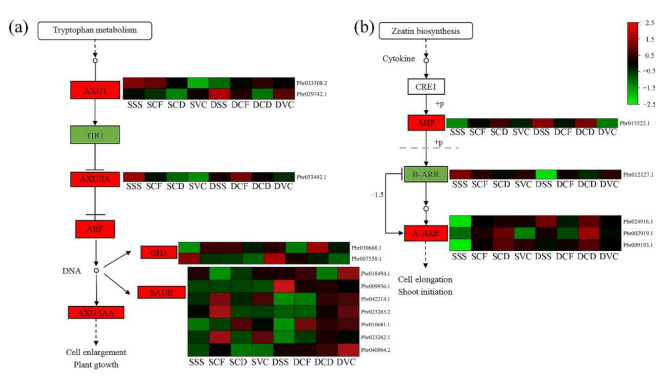
Heat maps of differentially expressed genes involved in phytohormone signaling transduction pathways: (**a**) auxin (AUX); (**b**) cytokinin (CTK). FPKM values are log_2_-based.

**Figure 5 plants-11-00580-f005:**
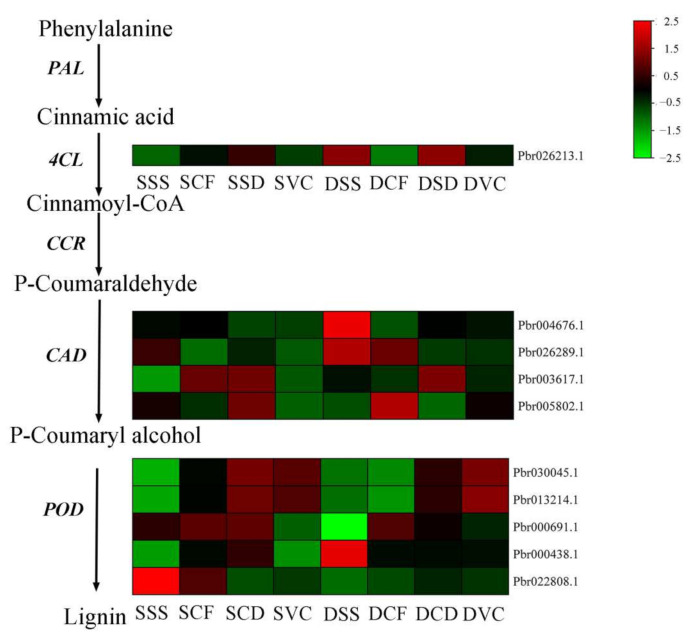
Heat maps of differentially expressed genes (DEGs) involved in phenylpropanoid biosynthesis following the three types of treatment. Heat maps show a transcriptional abundance of DEGs. FPKM values are log_2_-based.

**Figure 6 plants-11-00580-f006:**
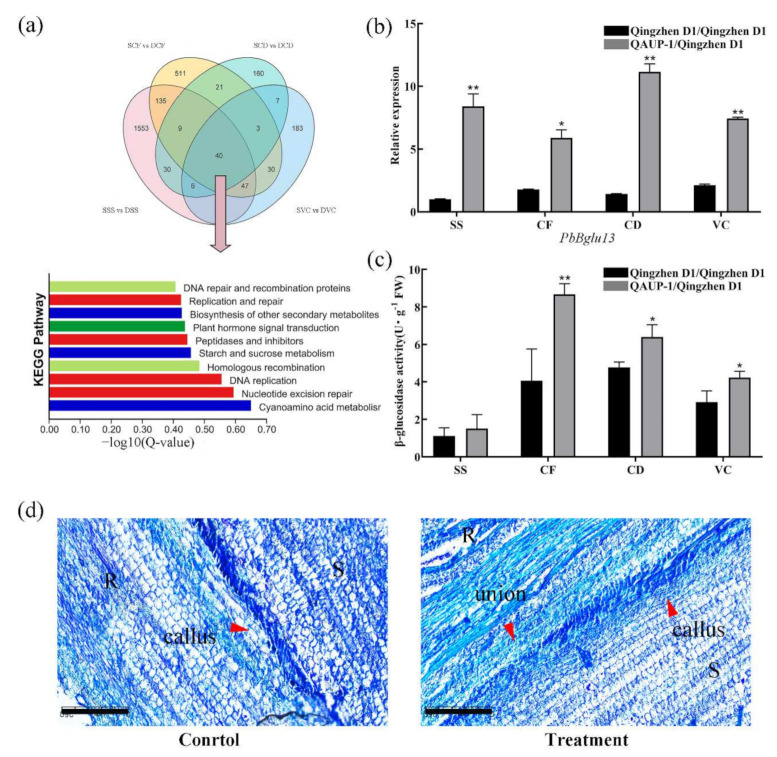
KEGG pathways enriched with differentially expressed genes (DEGs) and expression of a β−glucosidase gene, activity of β-glucosidase in different stages of homografts (“Qingzhen D1”/“Qingzhen D1”) and heterografts (“Qingzhen D1”/”QAUP-1”) of pear, and effect of β−glucosidase on graft union formation. (**a**) Venn diagram of DEGs in SSS vs. DSS, SCF vs. DCF, SCD vs. DCD, and SVC vs. DVC comparisons, as well as the 10 most significantly enriched KEGG pathways based on 40 DEGs. (**b**) Relative expression levels of the *PbBglu 13* gene and (**c**) activity of β-glucosidase in different stages after the grafting of homografts and heterografts. Error bars indicate SE (*n* = 3). Significant differences in expression or activity within a group are expressed as * *p* < 0.05 and ** *p* < 0.01. (**d**) Phenotypic observation of interface healing after β-glucosidase treatment. Scale bar = 250 μm.

**Figure 7 plants-11-00580-f007:**
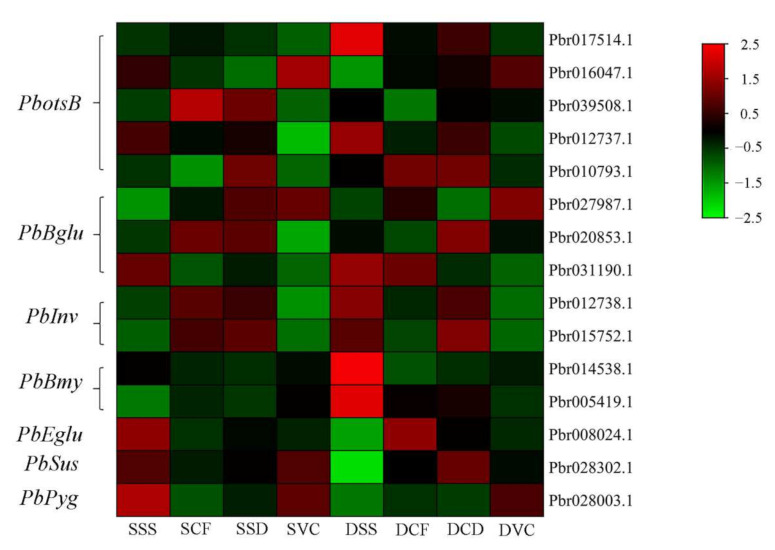
Heat maps of differentially expressed genes involved in the starch and sucrose metabolism pathways. FPKM values are log_2_−based.

**Figure 8 plants-11-00580-f008:**
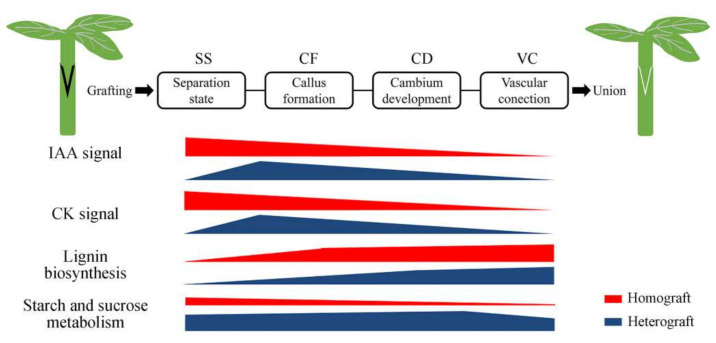
Model of graft union development in compatible and incompatible grafts of *Pyrus*.

**Table 1 plants-11-00580-t001:** The mineral element content of different stem parts of homografts and heterografts.

		Mineral Elements (mg·kg^−1^)					
		P	K	Mg	S	Ca	N
Above the grafting union	Homografts	629.45	1876.60	401.09	279.36	7422.08	1187.80
	Heterografts	755.16*	2062.01 *	473.18 *	290.96	7781.40	1224.04
At the grafting union	Homografts	825.01	2150.20	525.83	335.20	8255.29	1531.16
	Heterografts	821.81	2157.92	489.49	356.38	7199.10	1303.66
Below the grafting union	Homografts	976.54	2204.77	557.99 **	310.19	9723.24	1531.16
	Heterografts	945.31	2213.95	511.21	327.85 *	10159.86	1303.66

Values are the mean ± SE, *n* = 3. Significant differences within a group are expressed as * *p* < 0.05 and ** *p* < 0.01 (Student’s *t*-test).

## Data Availability

Not applicable.

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
