# Peer review of "Comparisons of Anatomical Characteristics and Transcriptomic Differences between Heterografts and Homografts in *Pyrus* L."

_plants, 2022, doi:10.3390/plants11050580_

Round 1
Reviewer 1 Report
This manuscript has a novel research direction and wide application value, but the description of the research purpose and significance is relatively lacking, and the coherence of each part of the article is not good. It is suggested to accept after minor modification as follows:
Line 3, Pyrus L.? The full Latin name of the plant should be used.
Line 14, Pear (Pyrus L.) The first occurrence of plant names should be marked with Latin names.
Line 15, “However, mechanisms of graft healing or incompatibility remain poorly understood.” for Pyrus or for all plants?
Line 26, Keywords should avoid duplication with words that appear in the title.
Line 28, The introduction should strengthen the description of the research purpose, Why Characteristics and Transcriptomic are selected for this study? And why heterograft and homograft are compared? Is homograft used as a control group? In addition, the significance of the research should also be highlighted in this part.
Line 304, varieties or Cultivars?
Line 305, Pyrus bretschneideri R: R is the initials of the person who names this plant? Should be “R.”.
Line 307, “In experiment I, one-year-old clonal plants of ‘Qingzhen D1’ and ‘QAUP-1’ planted in the Weifang Regional Experimental Park were used as scion and rootstock to be grafted. In experiment II, plantlets of ‘Qingzhen D1’ and ‘QAUP-1’ were used as scion and rootstock.” The description is not exact, which cultivar is the scion and which is the rootstock, scion and stock are identical, why are they divided into two experiments?
The first experiment was done inside the park and the second experiment was done outside the park?
one-year-old clonal plants and how old are plantlets? Why do comparisons of plants of different ages and sites need to be explained in the introduction part.
Line 310, “After grafting” means after the graft is healed?
Line 316, “Scion and rootstock diameters were measured one year after grafting.” Does it mean that the experimental plants were grown in MS medium for one year?
Line 304-313, Detailed grafting methods should be described in this section.
Line 329, The Binding Force During Graft Healing is better to list them separately.
Line 384, How many plants & how many samples as a repeat group? Should describe in detail.
Line 103, “silicon tubing collars” There is no prior description in the introduction and M&M part.
Line 105, “DAG” Full name should be marked when DAG first appears.
Figure 1. Adjust the layout and make the font bigger.
Table 1. Please use standard three-line tables.
“mg/kg” change to “μg·kg-1”.
The positive and negative error values can be removed when the data center is marked with significance analysis.
Figure 2. The electron microscope pictures are poor in resolution. It is recommended to adjust the layout and enlarge the size.
Figure 3. Completely unable to see the text content.
Line 235-239 should be in the introduction section.
In the discussion section, the future prospect of this research and the application value of the research results in practical production are lacking.
Author Response
Dear Reviewer,
Thank you very much for giving us the chance to revise our manuscript entitled “Comparisons of Anatomical Characteristics and Transcriptomic Differences between Heterografts and Homografts in Pyrus. L” (ID: We are grateful for your valuable suggestions for further improving the quality of our manuscript. We read the comments thoroughly and have revised the manuscript following your suggestions. Please find below our responses to the editor’s comments. We hope that the quality of revised version is improved.
Thank you very much again for your comments and suggestions. We are looking forward to hearing from you soon.
Yours sincerely
Dingli Li
2022.2.13

Reviewer 2 Report
This manuscript reported the anatomical and biochemical differences between n heterografts and homografts in Pyrus. The contents were interesting and were valuable for the readers. However, I have several concerns as indicated below.
- Provide complete biological and technical replication numbers.
- Provide photographs of both homograft and heterograft combinations showing one year status after grafting. It can be said that the combination of ‘QAUP-1/Qingzhen D1’ is incompatible combination or partly incompatible one?
- Two experimental designs (I and II) were set, but it was not clear which experimental design was applied to each analysis. I guess, anatomical analysis was based on the experimental design I, but how about other analyses? Show clearly.
- Lines 186-187, “A gene encoding β-glucosidase, PbBglu13, was more highly up-regulated in homografts than in heterografts in the healing process (Figure 6b)”. Also lines 285-286, “Acitivity of β-glucosidase was higher in homografts than heterografts (Figure 6c)”. These descriptions were right?
- The authors demonstrated the involvement of sugars in the graft healing process. Some potential sugar metabolizing-related genes could be found after transcriptional analysis? If so, related data should be added in the results and discussed.
Author Response
Dear Reviewer,
Thank you very much for giving us the chance to revise our manuscript entitled “Comparisons of Anatomical Characteristics and Transcriptomic Differences between Heterografts and Homografts in Pyrus. L” (ID: We are grateful for your valuable suggestions for further improving the quality of our manuscript. We read the comments thoroughly and have revised the manuscript following your suggestions. Please find below our responses to the editor’s comments. We hope that the quality of revised version is improved.
Thank you very much again for your comments and suggestions. We are looking forward to hearing from you soon.

Reviewer 3 Report
The manuscript describes anatomical and transcriptional changes in two graft unions (heterograft and homograft) that could improve our understanding of factors affecting graft compatibility of pears. However, I thought that the manuscript should be improved before being published in Plants for the following reasons.
- The Result section contains too many sentences that have already been or should be explained in the Introduction, Materials and Methods, and Discussion sections. Also, the authors should clearly organize the Results and Discussion sections to express the importance of the manuscript.
- You should specify the reference genome that was used for transcriptome analysis. In addition, the DEG names should be written in the manuscript by the Gene ID described in NCBI or database for the reference genome used in the experiment.
- The order of figures should be changed according to the results. The resolution of Figure 3 needs to be improved.
Author Response

(The authors gave the same response as above.)

Reviewer 4 Report
I reviewed your manuscript entitled "Comparisons of Anatomical Characteristics and Transcriptomic 2 Differences between Heterografts and Homografts in Pyrus", which you submitted to plants. I have evaluated the manuscript and listed my suggestions below. The study seems to share important inputs in graft healing in Heterografts and Homografts of Pyrus. but requires improvement before considering to publish.
The arrangement of the sections is not seen in the manuscript.
In lines 49-51 you mentioned that when pepper and tomato were grafted, parenchymatous callus formed at junctions of compatible and incompatible combinations. Why compatibility and incompatibility observed in the connection between these two plants?
As you know, most callus tissue is produced at the graft site by the rootstock. What caused the callus production to drop in heterograft? please explain more.
I also recommend you to do a better literature review about grafting to enrich introduction and discussion. For example you may address to the following papers:
Ebrahimi A and Vahdati K (2007) Improved success of Persian walnut grafting under environmentally controlled conditions. International Journal of Fruit Science. 6(4): 3-12.
Çoban, N., Öztürk, A. Determination of Graft Compatibility of Pear Cultivars Grafted on Different Rootstocks by Carbohydrate Analyses. Erwerbs-Obstbau (2022). https://doi.org/10.1007/s10341-021-00630-1
Rezaee R, Vahdati K, Grigoorian W, Valizadeh M (2008) Walnut grafting success and bleeding rate as affected by different grafting methods and seedling vigor. The Journal of Horticultural Science & Biotechnology. 83(1):94-99.
Author Response

(The authors gave the same response as above.)

Reviewer 5 Report
The manuscript entitled “Comparisons of anatomical characteristics and transcriptomic differences between heterografts and homografts in Pyrus” is in the scope of Plants. The manuscript provides good information on the differences in mechanisms of graft healing between homografts and heterografts.
- The manuscript needs major English editing
- In the Abstract section, the results and conclusion should be improved
- The general differences between homografts and heterografts should be improved in the introduction
- In line 30 please remove “(pp. 131–163)”
- In the material and methods the subtitle “2.3. Determination of Mineral Elements in Different Parts of Stem in Homografts and Heterografts” should be 4.3
- DAG is presented for the first time in line 105, so it should be presented complete expression “days after grafting (DAG)” and thereafter could be abbreviated
- Figure 1, 2 and Table 1 should be presented earlier to be close for related text
- Significance letters are presented in Table 1 although the used statistical analysis was Student’s t-test which is not correct
Author Response

(The authors gave the same response as above.)

Round 2
Reviewer 2 Report
I have satisfied the revised version; thereby, I have recommended publication.
Reviewer 3 Report
The revised manuscript was well corrected and improved enough to be accepted.
Reviewer 5 Report
The authors have adequately addressed all previous comments and the manuscript has been considerably improved.